# Plant Foliar Geometry as a Biomimetic Template for Antenna Design

**DOI:** 10.3390/biomimetics8070531

**Published:** 2023-11-07

**Authors:** Jose Ignacio Lozano, Marco A. Panduro, Rodrigo Méndez-Alonzo, Miguel A. Alonso-Arevalo, Roberto Conte, Alberto Reyna

**Affiliations:** 1Departamento de Electrónica y Telecomunicaciones, Centro de Investigación Científica y de Educación Superior de Ensenada (CICESE), Carretera Ensenada-Tijuana No. 3918, Zona Playitas, Ensenada 22860, Baja California, Mexico; joseilozano24@gmail.com (J.I.L.); aalonso@cicese.edu.mx (M.A.A.-A.); conte@cicese.mx (R.C.); 2Departamento de Biología de la Conservación, Centro de Investigación Científica y de Educación Superior de Ensenada (CICESE), Carretera Ensenada-Tijuana No. 3918, Zona Playitas, Ensenada 22860, Baja California, Mexico; mendezal@cicese.mx; 3Electrical and Electronic Engineering Department, Universidad Autónoma de Tamaulipas, UAMRR-R, Carretera Reynosa-San Fernando, Reynosa 88779, Tamaulipas, Mexico; alberto.reyna@docentes.uat.edu.mx

**Keywords:** antenna design, biomimetics, plant geometry, foliar structure, reflection coefficient, radiation pattern

## Abstract

Plant diversity includes over 300,000 species, and leaf structure is one of the main targets of selection, being highly variable in shape and size. On the other hand, the optimization of antenna design has no unique solution to satisfy the current range of applications. We analyzed the foliar geometries of 100 plant species and applied them as a biomimetic design template for microstrip patch antenna systems. From this set, a subset of seven species were further analyzed, including species from tropical and temperate forests across the phylogeny of the Angiosperms. Foliar geometry per species was processed by image processing analyses, and the resultant geometries were used in simulations of the reflection coefficients and the radiation patterns via finite differences methods. A value below −10 dB is set for the reflection coefficient to determine the operation frequencies of all antenna elements. All species showed between 3 and 15 operational frequencies, and four species had operational frequencies that included the 2.4 and 5 GHz bands. The reflection coefficients and the radiation patterns in most of the designs were equal or superior to those of conventional antennas, with several species showing multiband effects and omnidirectional radiation. We demonstrate that plant structures can be used as a biomimetic tool in designing microstrip antenna for a wide range of applications.

## 1. Introduction

Plants have been an inspiration source for the analysis and design of different systems in the physical and engineering fields. In plants, the reception, interception and exclusion of solar energy are critical steps, as plants receive energy in a wide frequency range, but are only able to utilize certain wavelengths in the electromagnetic spectrum to photosynthesize, mainly in the blue and red visible bands in the case of the chlorophyll pigments. Other values of wavelength are either intercepted through the accessory pigments, such as xantophylls and carotenoids, or discarded as reflected infrared radiation [1,2]. Evolution has strongly shaped the leaf form as a mainly flat and wide surface to intercept light and avoid self-shading [3]. The geometrical forms of leaves are strongly associated with their ecology and development, including light interception and water conduction [4]. In other words, a plant can be considered a natural wireless system. The geometrical forms of plant leaves have been studied in a wide range of wireless applications such as: energy harvesting [5,6], antenna designs [7,8], terahertz applications [9], Internet of Plants [10], thermal exchange [11], among others. Therefore, the structure of leaves can be employed as a biomimetic toolbox for a wide set of technological applications [12]. This should not be surprising, as there are over 300,000 described species of plants in the world, and it is estimated that an additional 150,000 are to be still described [13]. The foliar structure has been one of the main targets of diversification, and as such, the shape is influenced by the networks of water conduction [14,15], and the patterns of radiation capture [16,17,18].

One potentially critical application of the geometry of plant leaves is the design and production of antennas [19,20,21,22,23]. Previous work by Gielis et al., proposed the application of a super-formula to create many different foliar geometries [24,25,26]. These foliar geometries generated by the Gielis super-formula have provided an interesting ground for designing new types and forms of antennas, encompassing a wide range of applications. However, even when the Gielis super-formula can provide a mathematically feasible way to generate multilobated leaves, it is still difficult to emulate the vast diversity of leaf shapes and sizes in the natural world. Furthermore, other authors have employed single species to design antenna elemental geometry, by imitating biological structures [27] or bionic antennas [28]. Some examples include the simulation of the *Ginkgo biloba* leaf shape [29] and the banana leaf antenna [30].

Although the use of the natural geometries of plant leaves have been considered previously as a template for designing antenna elements, there is still an ample gap in our current capacity to find different geometrical configurations that outperform the state of art imposed by regular and conventional geometries. Therefore, this paper analyzes and studies the application of several novel geometries of plant leaves for designing antenna systems. First, we generated a database of the foliar geometry of 100 species, which was posteriorly refined to include the potentially best designs, that in our case, included seven different plant species to study and analyze in terms of simulated coefficient of reflection, and patterns of radiation: *Tetrapterys macrocarpa* (Malpighiaceae), *Sarcorhachis naranjoana* (Piperaceae), *Cissampelos owariensis* (Menispermaceae), *Paranomus spectrum* (Proteaceae), *Cesearia ilicifolia* (Salicaceae), *Liquidambar styraciflua* (Hamamelidaceae)*,* and *Quercus alba* (Fagaceae), a set of species that encompassed several clades of flowering plants across several biomes, including tropical and temperate forests and drylands. Therefore, the effect of the leaf geometric characteristics (elliptic, obovate, ovate, oblong, toothed, lobed leaves, pinnately lobed) was studied by analyzing these seven different types of leaves across species.

In conjunction with this, our main expectation is that some of the analyzed geometries would equal or even outperform the current structural designs of conventional antenna, and that some species might even find ground as templates for current technological applications [31]. Therefore, the main differences in our work with respect to existing work are: the development of a methodology to generate antenna designs considering the natural geometry of plant leaves and secondly, a demonstration via performance analyses, that new antenna geometries can be achieved using foliar geometries as templates that can be quantified in terms of the reflection coefficient and the radiation pattern of each antenna structure. We further show, via simulation results, that the performance of each plant species’ leaf design using a CST electromagnetic solver, indicates possible new applications for developing specific uses for different types of antennas through a wide set of specific wavelengths.

## 2. Materials and Methods

Two important elements are required to generate the antenna design: software to process images (SolidWorks) of the leaves’ geometries and the electromagnetic solver to assess the design models (CST Microwave Studio) [32].

Figure 1 illustrates the method of the design process to generate the radiating elements using the image of the leaf geometry. The geometry of the leaves is generated by processing the images of the chemically cleared plant leaves as described in [33]. The foliar geometry of 100 species is generated and analyzed. These 100 species are shown in Table 1 using the details provided by [33] as family and type of leaf. The antenna design methodology using the image or template of the leaf geometry follows the next steps (Figure 1):The image of the leaf geometry (for each species in Table 1) is taken from the Manual of Leaf Architecture [33]. This is in order to obtain a wide range of leaf images. Then, we take the image set in this leaf manual by considering the front view and knowing the name and family of each leaf architecture. Furthermore, this leaf manual provides an extensive selection of many kinds of leaf architectures found in nature.The original leaf image (taken from nature) is converted to black and white. The black and white image can be generated using color filter software.CAD design software (for example SolidWorks) is used to generate the image. SolidWorks processes the image in black and white. The function “Sketch Picture” of SolidWorks can be used to detect the black and white leaf geometry. Then, a collection of samples or points is generated to provide a tridimensional model by using a compatible format with CST solver.The processed images are exported to the CST electromagnetic solver to set dimensions and substrate properties. The substrate FR4 is considered in each design geometry with the next electrical properties: dimensions of 100 mm × 100 mm, a thickness of 1.6 mm, relative permittivity *εr* = 4.2 and tangent loss of 0.025.Each leaf geometry of Table 1 is characterized in the CST solver. The CST software uses the method of finite differences in the time domain (FDTD) to provide the radiation pattern and the reflection coefficient of each leaf geometry. Different radiation pattern cuts (horizontal and vertical) can be extracted from the 3D pattern obtained.

Basically, this methodology is applied to consider the basic geometric characteristics of the plant leaves to design new antennas and investigate the effects on performance of the geometry. As shown in Table 1, there are many geometric characteristics found in the plants’ leaves such as: elliptic, obovate, ovate, circular base, toothed, lobed leaves, pinnately lobed, among others. It would be interesting to see the design impact of each different geometric characteristic of the plants’ leaves. The behavior of these leaf geometries can be studied as antenna systems. The idea is to use the geometry of the plant leaves illustrated in Table 1 to design new antenna structures.

More sophisticated methods can be used to generate high resolution images that could be used to analyze the veins of the plant leaves. However, that can be performed as an extension of this work. The design results for the leaf geometries are illustrated in the next section.

## 3. Results

The plant leaf geometries illustrated previously in Table 1 were simulated in the CST electromagnetic solver to be analyzed as antenna systems. The reflection coefficient (*S*_11_) is used to determine the operation frequencies of each leaf antenna design. Basically, a value of −10 dB (or below) is set for *S*_11_ to determine the operation frequencies of all antenna elements. Then, the electromagnetic radiation can be determined for all the operation frequencies or some frequencies of interest.

Table 2 shows the frequency values to be operated by the 100 antennas generated by the leaf geometries (illustrated in Table 1). As mentioned previously, the operation frequency is determined using the reflection coefficient (S_11_ < −10 dB).

Table 2 considers frequency values between 600 MHz and 11 GHz. This table can be useful for the antenna designers to identify which antenna elements could operate in certain applications by considering the operation frequency. CST Microwave Studio applies the method of FDTD to assess the antenna structures [32]. This method is very rigorous at characterizing the design of each antenna. Therefore, these frequency values (shown in Table 2) can be considered a good approximation in practical terms.

Table 3, Table 4, Table 5, Table 6 and Table 7 illustrate a classification of frequency bands to be operated by the antenna elements according to the electromagnetic spectrum. As shown in these tables, many antenna elements show multiband operation. This classification is very useful because these frequency bands are employed in applications of radar, terrestrial and satellite communications. The frequency bands to be operated by more antenna elements are the C and X bands. Furthermore, the frequency band operated by fewer antenna elements is the L band. This frequency band has a smaller frequency range than the other bands.

Each antenna element could be worked and analyzed to operate at a specific frequency. However, this analysis of operation frequencies helps us to identify what kind of leaf design geometries can adapt in a better way to a set frequency range. Some modern applications or bands of interest can be included in this analysis. For example, Table 8 illustrates the antenna elements that can operate at frequencies of Wi-Fi (2.4 GHz) and 5 GHz. These frequencies are often used in wireless technologies.

As illustrated previously, each leaf geometry presents a different geometric shape. Therefore, we selected seven particular leaves in order to illustrate a different geometrical shape: *Tetrapterys macrocarpa* (Malpighiaceae), *Sarcorhachis naranjoana* (Piperaceae), *Cissampelos owariensis* (Menispermaceae), *Paranomus spectrum* (Proteaceae), *Cesearia ilicifolia* (Salicaceae), *Liquidambar styraciflua* (Hamamelidaceae)*,* and *Quercus alba* (Fagaceae). The effect of the leaf’s geometric characteristics (elliptic, obovate, ovate, oblong, toothed, lobed leaves, pinnately lobed) can be studied by analyzing these seven different types of leaves across species.

Figure 2, Figure 3, Figure 4, Figure 5, Figure 6, Figure 7 and Figure 8 illustrate the antenna design in CST and the results obtained for the seven leaf geometries selected. These figures show the behavior of the reflection coefficient (defined as *S*_11_), the design configuration in the CST electromagnetic solver, 3D electromagnetic radiation and some cuts of the radiation pattern (vertical and horizontal) for different values of frequency of each plant leaf geometry. As shown in these figures, it is interesting to note that all the design cases present multiple operation frequencies. All the cases generate a multiband response in the behavior of the reflection. Therefore, the design impact of each plant leaf will be described, providing details of each different leaf design to give a more detailed description.

The case of elliptic geometry (*Tetrapterys macrocarpa* Malpighiaceae, Figure 2) provides six frequencies of operation below 5 GHz (0.87 GHz, 1.62 GHz, 2.19 GHz, 2.61 GHz, 3.3 GHz and 4.11 GHz), another operation frequency is obtained at 9.2 GHz and for frequencies higher than ≈13 GHz the reflection coefficient (Figure 2b) remains below −10 dB. The obtained radiation for the frequencies of 0.87 GHz and 3.3 GHz is illustrated in Figure 2d–g. The horizontal and vertical cut of the radiation pattern obtained for each frequency is illustrated. As observed in these figures, different radiation characteristics are obtained for different values of frequency. Better radiation characteristics are obtained for 0.87 GHz (radiation is obtained in almost half of all the cuts or plane). However, desirable radiation characteristics persist when the frequency is increased up to 3.3 GHz.

The ovate case (*Sarcorhachis naranjoana* Piperaceae, Figure 3) generates at least eight interesting frequency bands below 10 GHz (1.05 GHz, 2.9–3.6 GHz, 4.07–4.28 GHz, 4.56–4.71 GHz, 5.15–5.32 GHz, 6.42–6.6 GHz, 7.08–7.22 GHz, 8.4–8.655 GHz, 9.01–9.13 GHz, 9.59–9.76 GHz). Furthermore, the reflection coefficient (Figure 3b) remains below −10 dB for frequencies higher than 11.6 GHz. The radiation pattern cuts (vertical and horizontal) at *f* = 1.05 GHz and *f* = 3.09 GHz are illustrated in Figure 3d–g. As illustrated in Figure 3d–g, good radiation characteristics are obtained for these frequency cuts. The ovate geometry impacts the generation of more design frequencies below 10 GHz with respect to elliptic geometry.

The circular base geometry (*Cissampelos owariensis* Menispermaceae, Figure 4) provides at least seven interesting frequency bands below 10 GHz (1.1–1.34 GHz, 1.61–1.81 GHz, 2.4–2.55 GHz, 2.98–3.11 GHz, 3.75–3.98 GHz, 6.05–6.40 GHz and 8.4–8.96 GHz). In addition, the reflection coefficient (Figure 4b) remains below −10 dB for frequencies higher than ≈14.9 GHz. The pattern cuts (vertical and horizontal) at *f* = 1.2 GHz and *f* = 6.27 GHz are illustrated in Figure 4d–g. The leaf width impacts the generation of better radiation characteristics (there is more radiation in all the cuts) with respect to the previous cases. This phenomenon can be observed through the obtained radiation patterns at the operating frequencies of 1.2 GHz and 6.27 GHz.

The obovate case (*Paranomus spectrum* Proteaceae, Figure 5) provides several frequency bands below 10 GHz: 1.5–1.66 GHz, 2.19–2.92 GHz, 5.1–5.47 GHz, 5.82–6.1 GHz, 6.28–6.75 GHz, 7.28–7.91 GHz, 8.36–8.92 GHz, 9.27–10 GHz. This case generates the best behavior for high frequencies (with respect to the other cases), i.e., for frequencies higher than ≈9.27 GHz the reflection coefficient (Figure 5b) remains below −10 dB. An exciting observation can be made from the radiation patterns (observe Figure 5d–g) obtained at *f* = 2.34 GHz and *f* = 5.31 GHz. These figures exhibit improved radiation characteristics for higher frequency values.

The toothed case (*Cesearia ilicifolia* Salicaceae, Figure 6) generates more operation frequency bands below 10 GHz with respect to the other cases (0.8–0.97 GHz, 1.62–1.81 GHz, 2.36–2.56 GHz, 3.22–3.38 GHz, 4.55–4.74 GHz, 5.81–6.03 GHz, 6.4–7.6 GHz, 8.07–8.47 GHz, 8.83–9.15 GHz and 9.6–9.92 GHz). Furthermore, the reflection coefficient (Figure 6b) remains below −10 dB for frequencies higher than 18 GHz. Figure 6d–g illustrates good radiation characteristics considering the horizontal and vertical cuts for the frequency values at *f* = 5.91 GHz and *f* = 6.51 GHz.

The geometry of lobed leaves (*Liquidambar styraciflua* Hamamelidaceae, Figure 7) provided seven operation frequency bands below 10 GHz (1.98–2.07 GHz, 4.11–4.34 GHz, 5.46–5.76 GHz, 5.86–6.17 GHz, 7.24–7.52 GHz, 7.79–8.485 GHz and 8.99–9.84 GHz). Some of these frequency bands presented values of *S*_11_ (Figure 7b) very close to −10 dB. Figure 7d,e illustrates the obtained radiation cuts (horizontal and vertical) at *f* = 2.34 GHz and *f* = 5.31 GHz. The depicted figures demonstrate notable radiation characteristics at the respective frequency values.

The geometry of pinnately lobed (*Quercus alba* Fagaceae, Figure 8) provided several frequency bands and the widest frequency band below 10 GHz with respect to the other cases (2.41–2.57 GHz, 3.1–3.3 GHz, 3.66–3.95 GHz, 4.56–4.68 GHz, 5.05–6.13 GHz, 7.47–9.801 GHz). The reflection coefficient (Figure 8b) illustrates a frequency band of ≈2.2 GHz (from 7.47 GHz to 9.8 GHz). Furthermore, the horizontal and vertical cuts of obtained radiation for *f* = 3.18 GHz and *f* = 5.88 GHz are shown in Figure 8d–g. A better behavior of the radiation characteristics for the higher frequency value is illustrated.

## 4. Discussion

By exploring a wide set of geometries derived from 100 species, we were able to distinguish features that provided a flowchart to select geometries apt to satisfy technological applications. In particular, the subset of seven species offers a template to deepen our comprehension of the design requirement involved in biomimetic antenna design. This work, to our knowledge, is a comprehensive effort to study plant biomimetic antenna designs, and the simulations performed in CST stimulate further research, given the excellent properties that certain leaf architectures provide in comparison to the conventional geometries currently employed.

For example, Table 9 illustrates a comparative analysis of the performance of each plant leaf design. This comparative analysis is achieved in terms of the leaf geometry, design impact and operation frequencies below 10 GHz. Although more than one geometry could contribute to the same design impact (or similar performance characteristics), several interesting design characteristics can be considered as a result of this study: the obovate geometry presented the best behavior for high frequencies, the toothed case provided more frequency bands below 10 GHz, the circular base geometry (the widest plant leaf) generated the best radiation characteristics and the pinnately lobed case provided the widest frequency band.

Some limitations of this work and potential future work can be extended from the current study as follows:Each leaf architecture (in this work) considers vein characteristics of the first order. Therefore, more vein characteristics (of higher order) can be considered in order to obtain better performance features.This study only considers the response of the leaf geometry found in nature. It could be interesting to try to generate a geometry that considers the optimization of the design parameters in order to improve performance. Furthermore, some of the interesting characteristics found in this study could be used (or mixed) to try to create new design geometries or super-geometries. Therefore, we consider that future work searching the optimization of biomimetic designs based on plant leaf architectures will further support the development of specific applications in antenna design.

## 5. Conclusions

The foliar geometries of 100 plant species were analyzed and applied as a biomimetic design template for patch antenna systems. The foliar geometry per species was processed through image processing analyses, and the resultant geometries were used in simulations of the reflection coefficients and the radiation patterns via the method of FDTD. Furthermore, the effect of the leaf geometric characteristics (elliptic, obovate, ovate, oblong, toothed, lobed leaves, pinnately lobed) was analyzed to study the design impact. This impact study was achieved by selecting and analyzing seven species from tropical and temperate forests across the phylogeny of the Angiosperms.

The obovate geometry presented the best behavior for high frequencies. The toothed case provided more frequency bands below 10 GHz. The circular base geometry (the widest plant leaf) generated the best radiation characteristics, and the pinnately lobed case provided the widest frequency band. All the design cases presented multiple operation frequencies. All the design cases generated a multiband response in the behavior of the reflection coefficient. All species showed between 3 and 15 operational frequencies, and four species had operational frequencies that included the 2.4 and 5 GHz bands. The reflection coefficients and the radiation patterns in most of the designs were equal or superior to those of conventional antennas, with several species showing multiband effects and omnidirectional radiation.

## Figures and Tables

**Figure 1 biomimetics-08-00531-f001:**
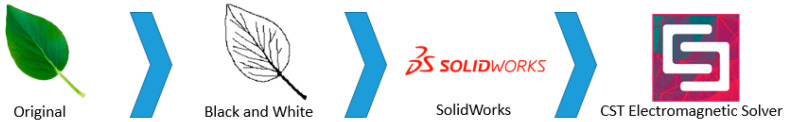
Basic method to assess the structure or each leaf design as a radiating element.

**Figure 2 biomimetics-08-00531-f002:**
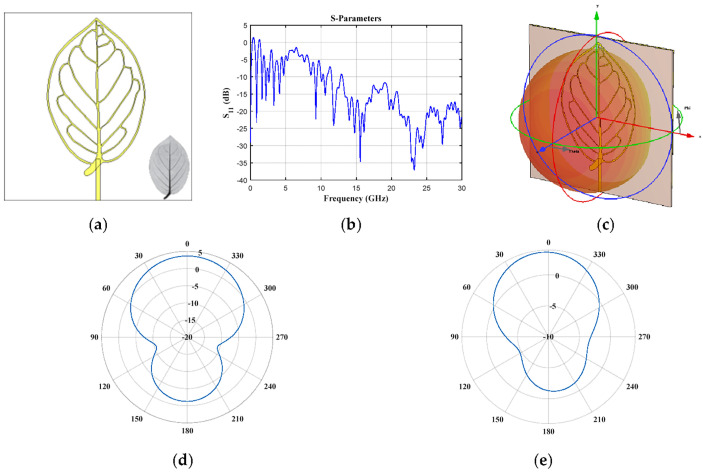
*Tetrapterys macrocarpa* Malpighiaceae (elliptic) (**a**) front view, (**b**) *S*_11_ parameter, (**c**) 3D pattern at 0.87 GHz, and cuts of the radiation pattern at *f* = 0.87 GHz (**d**) vertical and (**e**) horizontal cut, and at *f* = 3.3 GHz (**f**) vertical and (**g**) horizontal cut.

**Figure 3 biomimetics-08-00531-f003:**
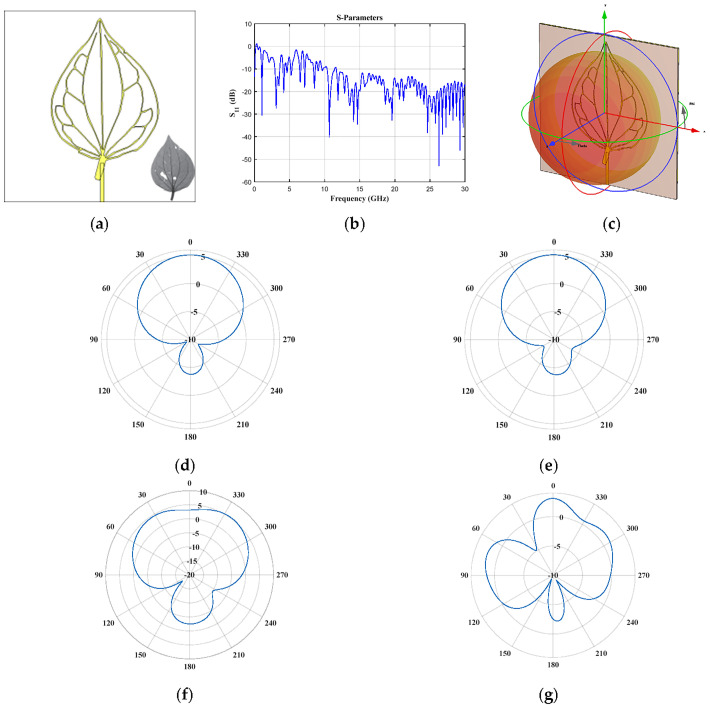
*Sarcorhachis naranjoana* Piperaceae (ovate) (**a**) front view, (**b**) *S*_11_ parameter, (**c**) 3D pattern at 1.05 GHz, and cuts of the radiation pattern at *f* = 1.05 GHz (**d**) vertical and (**e**) horizontal cut, and at *f* = 3.09 GHz (**f**) vertical and (**g**) horizontal cut.

**Figure 4 biomimetics-08-00531-f004:**
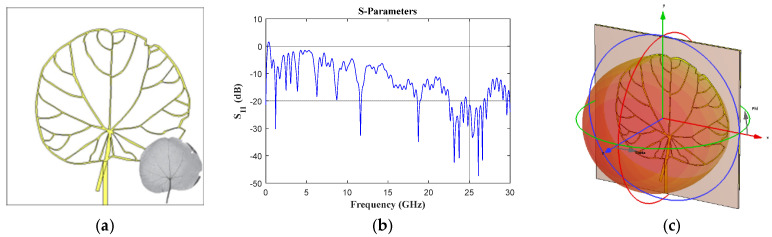
*Cissampelos owariensis* Menispermaceae (circular base) (**a**) front view, (**b**) *S*_11_ Parameter, (**c**) 3D pattern at 1.2 GHz, and cuts of the radiation pattern at *f* = 1.2 GHz (**d**) vertical and (**e**) horizontal cut, and at *f* = 6.27 GHz (**f**) vertical and (**g**) horizontal cut.

**Figure 5 biomimetics-08-00531-f005:**
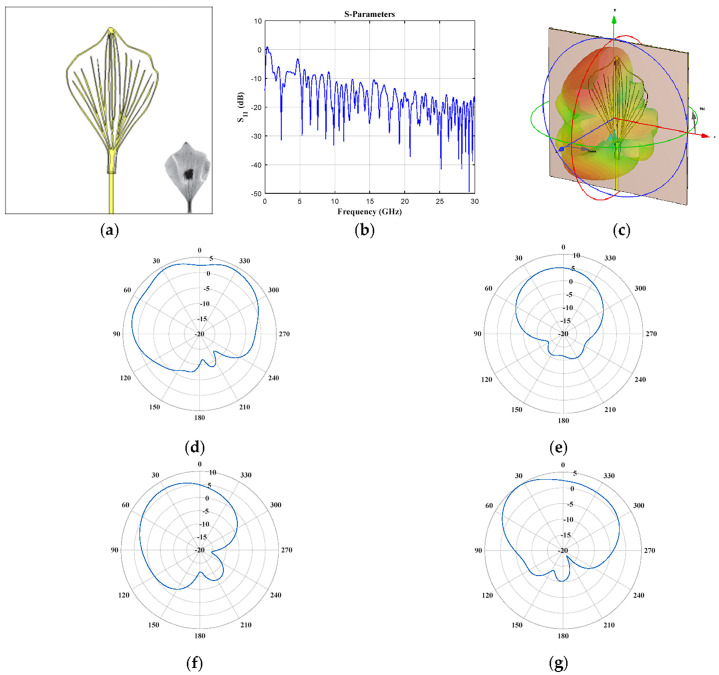
*Paranomus spectrum* Proteaceae (obovate) (**a**) Front view, (**b**) *S*_11_ Parameter, (**c**) 3D radiation pattern at 2.34 GHz, and cuts of the radiation pattern at *f* = 2.34 GHz (**d**) vertical and (**e**) horizontal cut, and at *f* = 5.31 GHz (**f**) vertical and (**g**) horizontal cut.

**Figure 6 biomimetics-08-00531-f006:**
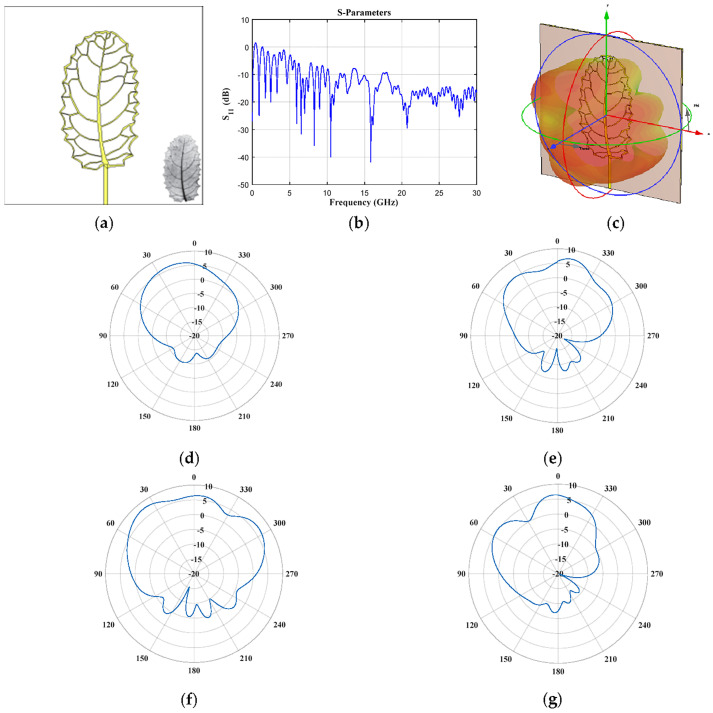
*Cesearia ilicifolia* Salicaceae (toothed) (**a**) Front view, (**b**) *S*_11_ Parameter, (**c**) 3D radiation pattern at 5.91 GHz, and cuts of the radiation pattern at *f* = 5.91 GHz (**d**) vertical and (**e**) horizontal cut, and at *f* = 6.51 GHz (**f**) vertical and (**g**) horizontal cut.

**Figure 7 biomimetics-08-00531-f007:**
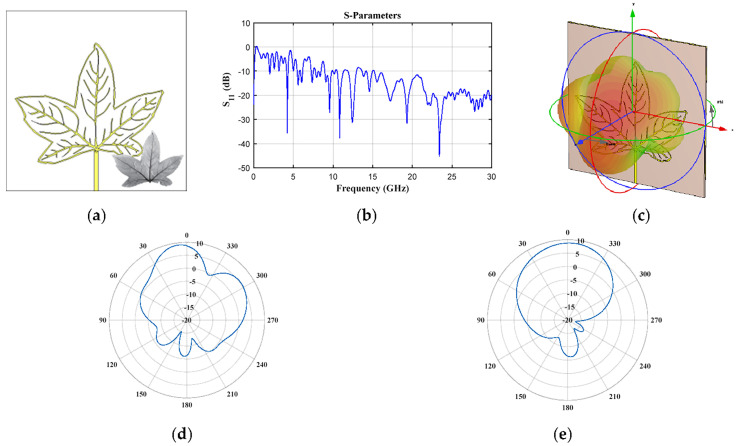
*Liquidambar styraciflua* Hamamelidaceae (lobed leaves) (**a**) front view, (**b**) *S*_11_ Parameter, (**c**) 3D pattern at 4.23 GHz, and cuts of the pattern at *f* = 4.23 GHz (**d**) vertical and (**e**) horizontal cut, and at *f* = 5.58 GHz (**f**) vertical and (**g**) horizontal cut.

**Figure 8 biomimetics-08-00531-f008:**
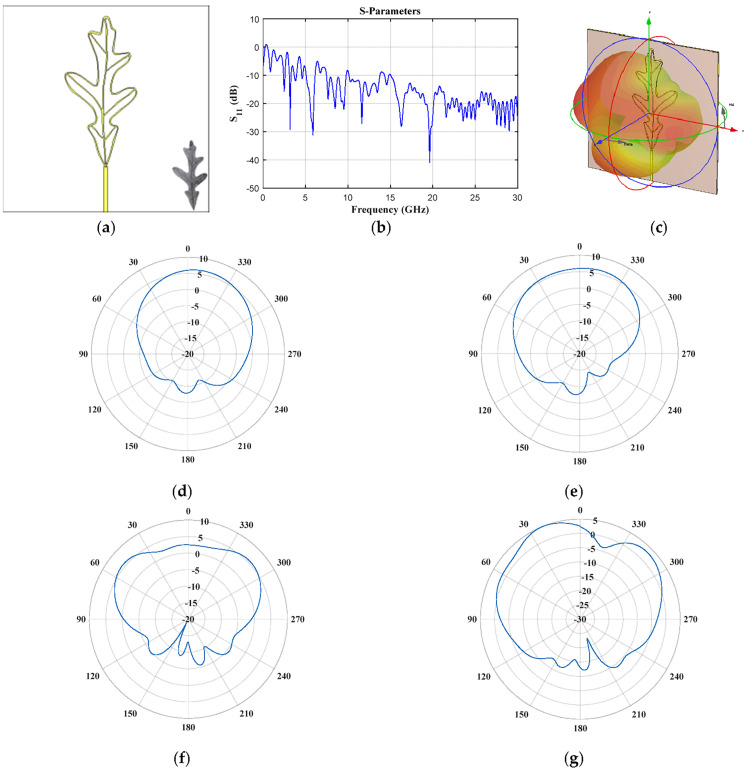
*Quercus alba* Fagaceae (pinnately lobed) (**a**) front view, (**b**) *S*_11_ Parameter, (**c**) 3D radiation pattern at 3.18 GHz, and cuts of the pattern at *f* = 3.18 GHz (**d**) vertical and (**e**) horizontal cut, and at *f* = 5.88 GHz (**f**) vertical and (**g**) horizontal cut.

**Table 1 biomimetics-08-00531-t001:** Set of species analyzed and assessed as antenna designs.

1	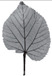	*Tillia Mandshurica*Malvaceae	11	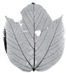	*Acer franchetii* Sapindaceae
2	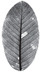	*Davilla rugosa* Dilleniaceae	12	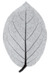	*Tetrapterys macrocarpa* Malpighiaceae
3	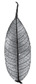	*Stemonoporus nitidus* Dipterocarpaceae	13	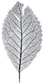	*Eucryphia glutinosa* Cunoniaceae
4	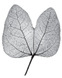	*Bauhinia madagascariensis* Fabaceae	14	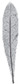	*Licania michauxii* Chrysobalanaceae
5	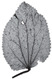	*Tetracentron sinense* Trochodendraceae	15	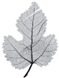	Morus microphylla Moraceae
6	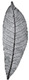	*Buchanania arborescens* Anacardiaceae	16	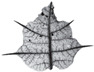	*Comocladia dodonaea* Anacardiaceae
7	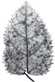	*Aristotelia racemosa* Elaeocarpaceae	17	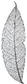	*Sorindeia gilletii* Anacardiaceae
8	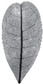	*Bombacopsis rupicola* Malvaceae	18	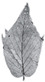	*Leepierceia preartocarpoides* Proteales
9	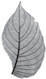	*Rhynchoglossum azureum* Gesneriaceae	19	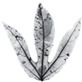	*Adenia heterophylla* Passifloraceae
10	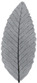	*Nothofagus procera* Nothofagaceae	20	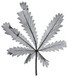	*Potentilla recta*Rosaceae
21	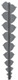	*Dryandra longifolia* Proteaceae	31	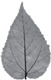	*Carrierea calycina* Salicaceae
22	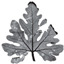	*Cucurbita cylindrata* Cucurbitaceae	32	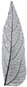	*Dalechampia cissifolia* Euphorbiaceae
23	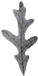	*Quercus alba*Fagaceae	33	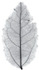	*Croton hircinus* Euphorbiaceae
24	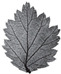	*Rubus mesogaeus* Rosaceae	34	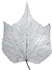	*Dombeya elegans* Malvaceae
25	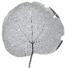	*Cissampelos owariensis* Menispermaceae	35	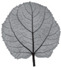	*Tetrameles nudiflora* Datiscaceae
26	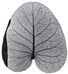	*Phyllanthus poumensis* Phyllanthaceae	36	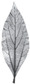	*Phoebe costaricana* Lauraceae
27	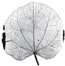	*Cercidiphyllum japonicum* Cercidiphyllaceae	37	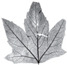	*Platanus racemosa* Platanaceae
28	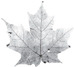	*Acer saccharinum* Sapindaceae	38	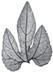	*Trichosanthes formosana* Cucurbitaceae
29	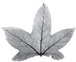	*Liquidambar styraciflua* Hamamelidaceae	39	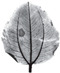	*Paliurus ramosissimus* Rhamnaceae
30	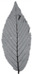	*Ostrya guatemalensis* Betulaceae	40	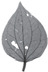	*Sarcorhachis naranjoana* Piperaceae
41	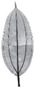	*Topobea watsonii* Melastomataceae	51	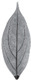	*Cleistanthus oligophlebius* Phyllanthaceae
42	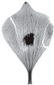	*Paranomus sceptrum* Proteaceae	52	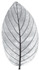	*Rhamnidium elaeocarpum* Rhamnaceae
43	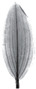	*Potamogeton amplifolius* Potamogetonaceae	53	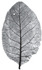	*Cotinus obovatus* Anacardiaceae
44	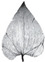	*Maianthemum dilatatum* Ruscaceae	54	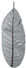	*Santiria samarensis* Burseraceae
45	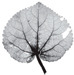	*Cercidiphyllum japonicum* Cercidiphyllaceae	55	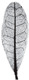	*Aextoxicon punctatum* Aextoxicaceae
46	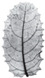	*Casearia ilicifolia*Salicaceae	56	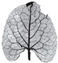	*Antigonon cinerascens* Polygonaceae
47	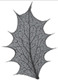	*Mahonia wilcoxii* Berberidaceae	57	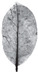	*Capsicodendron pimenteira* Canellaceae
48	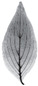	*Cornus officinalis* Cornaceae	58	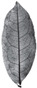	*Tapura guianensis* Dichapetalaceae
49	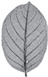	*Isoptera lissophylla* Dipterocarpaceae	59	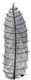	*Comocladia glabra* Anacardiaceae
50	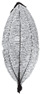	*Tococa aristata* Melastomataceae	60	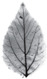	*Viburnum setigerum* Adoxaceae
61	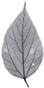	*Bixa orellana* Bixaceae	71	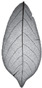	*Popowia congensis* Annonaceae
62	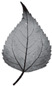	*Philactis zinnioides* Asteraceae	72	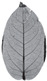	*Banisteriopsis laevifolia* Malpighiaceae
63	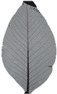	*Vitex limonifolia* Lamiaceae	73	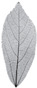	*Microcos tomentosa* Malvaceae
64	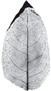	*Kermadecia sinuata* Proteaceae	74	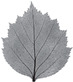	*Crataegus brainerdii* Rosaceae
65	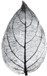	*Glochidion bracteatum* Phyllanthaceae	75	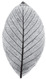	*Tetracera podotricha* Dilleniaceae
66	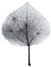	*Populus jackii* Salicaceae	76	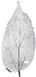	*Celtis cerasifera* Cannabaceae
67	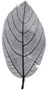	*Apeiba macropetala* Malvaceae	77	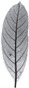	*Guarea tuberculata* Meliaceae
68	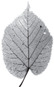	*Tilia heterophylla* Malvaceae	78	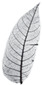	*Cedrela angustifolia* Meliaceae
69	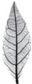	*Alchornea polyantha* Euphorbiaceae	79	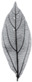	*Diospyros maritima* Ebenaceae
70	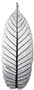	*Pseudolmedia laevis* Moraceae	80	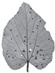	*Eriolaena malvacea* Malvaceae
81	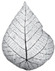	*Macaranga bicolor* Euphorbiaceae	91	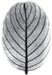	*Tetracera rotundifolia* Dilleniaceae
82	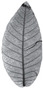	*Odontadenia geminata* Apocynaceae	92	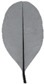	*Calophyllum calaba*Clusiaceae
83	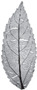	*Flacourtia rukam*Salicaceae	93	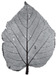	*Cissus caesia*Vitaceae
84	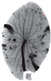	*Brasenia schreberi* Cabombaceae	94	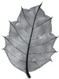	*Desfontainea spinosa* Desfontaineaceae
85	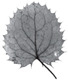	*Mahoberberis neubertii* Berberidaceae	95	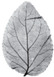	*Aphaerema spicata*Salicaceae
86	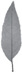	*Fraxinus floribunda* Oleaceae	96	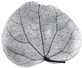	*Cyclea merrillii* Menispermaceae
87	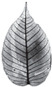	*Parinari campestris* Chrysobalanaceae	97	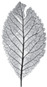	*Eucryphia glandulosa* Cunoniaceae
88	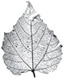	*Melanolepis multiglandulosa* Euphorbiaceae	98	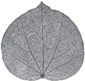	*Diploclisia kunstleri* Menispermaceae
89	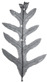	*Stenocarpus sinuatus* Proteaceae	99	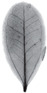	*Buxus glomerata* Buxaceae
90	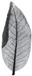	*Couepia paraensis* Chrysobalanaceae	100	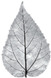	*Croton hircinus* Euphorbiaceae

**Table 2 biomimetics-08-00531-t002:** Frequency values (in GHz) to be operated by the 100 antennas generated by the leaf geometries of 100 species (Spp.) illustrated in Table 1.

Spp.	Frequency Values (GHz)
**1**	1.26, 1.95, 3.06, 3.72, 4.44, 5.58, 6.18, 7.23, 7.83, 8.97, 9.54, 10.77
**2**	1.05, 1.98, 3.24, 4.23, 5.25, 6.15, 7.2, 8.37, 9.18, 10.26
**3**	0.99, 1.29, 2.25, 3.36, 3.9, 5.61, 7.5, 8.07, 8.46, 9.03, 10.08
**4**	2.01, 7.71, 9.72
**5**	1.02, 2.22, 5.52, 6, 6.78, 7.59
**6**	4.53, 5.16, 7.95, 8.55, 9.06, 10.29
**7**	0.84, 3.09, 4.38, 5.1, 5.73, 6.75
**8**	2.67, 4.23, 4.98, 5.61, 7.08, 8.61
**9**	0.78, 1.44, 2.01, 2.46, 3.12, 4.05, 4.41, 5.28, 6.63, 7.17, 8.97, 10.8
**10**	0.87, 1.62, 2.4, 3.06, 3.63, 4.17, 5.76, 6.57, 7.26, 7.92, 8.46, 8.91, 9.54, 10.17
**11**	2.16, 2.7, 6.06, 7.05, 9.72, 10.92
**12**	0.87, 1.62, 2.19, 2.61, 3.3, 4.11, 9.27
**13**	0.87, 1.59, 2.34, 3.15, 4.44, 5.82, 6.51, 7.5, 7.89, 9.24, 10.02
**14**	4.53, 4.89, 5.61, 8.49, 9.66
**15**	0.72, 1.98, 2.61, 3.51, 4.23, 5.34, 5.91, 6.9, 7.56, 8.22, 8.97, 9.72
**16**	0.84, 1.5, 3.66, 4.68, 6.06, 6.72, 8.43, 9.57, 9.99
**17**	2.07, 2.64, 3.15, 4.26, 4.92, 5.46, 6, 7.38, 8.13, 9.09, 9.78
**18**	0.78, 1.47, 2.07, 2.76, 3.81, 4.92, 6.57, 7.68, 8.4, 9.21, 10.17
**19**	1.14, 2.67, 3.54, 4.68, 6.03, 6.51, 7.08, 7.65, 8.25, 9.54
**20**	1.11, 1.77, 3.12, 5.7, 7.74, 8.85, 9.72
**21**	8.88, 10.47
**22**	2.16, 2.88, 5.28, 6.33, 6.96, 7.56, 9, 9.84
**23**	2.49, 3.18, 3.84, 4.62, 5.88, 7.65, 8.49, 9.54
**24**	0.75, 1.41, 2.16, 3.39, 4.32, 4.77, 6.69, 7.44, 7.95, 8.61, 9.66, 10.38
**25**	1.2, 1.71, 2.49, 3.06, 3.87, 6.27, 8.73, 11.61
**26**	1.98, 2.49, 2.94, 3.48, 3.9, 4.53, 5.04, 5.52, 6.48, 7.08, 8.31, 9, 9.813731
**27**	1.14, 2.13, 2.91, 3.63, 7.23, 7.59, 8.19, 8.79, 9.27, 9.96
**28**	0.96, 2.43, 2.97, 3.36, 4.17, 5.4, 7.41, 10.77
**29**	2.01, 4.23, 5.58, 6.06, 7.35, 7.92, 8.37, 9.09, 9.57, 10.83
**30**	0.96, 2.19, 3.45, 4.08, 5.73, 7.32, 7.98, 8.55, 9.09, 9.66
**31**	2.07, 2.46, 2.91, 4.95, 5.67, 6.54, 7.47, 8.52, 9.3, 10.17
**32**	5.22, 6.39, 7.29, 8.16, 9.45, 10.41, 10.95
**33**	3.06, 4.08, 5.04, 6.93, 7.41, 8.16, 9.3, 9.99
**34**	0.96, 3.12, 5.7, 9.57
**35**	0.72, 2.79, 4.8, 5.55, 6.06, 6.69, 7.71, 9.21, 9.66, 10.38
**36**	0.99, 1.62, 2.07, 5.67, 6.48, 7.44, 8.13, 8.94
**37**	2.19, 3.75, 4.23, 4.77, 5.34, 5.82, 6.42, 7.17, 8.73, 9.63, 10.47
**38**	1.62, 4.98, 5.73, 6.03, 7.38, 8.97, 9.75
**39**	1.26, 2.34, 3.24, 3.63, 4.65, 5.43, 6.75, 7.26, 7.8, 8.36, 8.91, 9.51, 10.02
**40**	1.05, 3.09, 3.48, 4.17, 4.65, 5.22, 6.51, 7.14, 8.55, 9.06, 9.66, 10.65
**41**	2.97, 4.26, 5.46, 6.6, 7.83, 9.12
**42**	1.59, 2.34, 2.76, 5.31, 5.94, 6.48, 7.56, 8.7, 9.84
**43**	1.32, 2.55, 6.12, 7.35, 8.55, 9.54
**44**	1.29, 2.7, 3.9, 4.95, 6.36, 7.59, 8.31, 9, 9.57, 10.95
**45**	1.71, 6.75, 7.5, 8.19, 8.91
**46**	0.9, 1.74, 2.46, 3.3, 4.62, 5.91, 6.51, 6.99, 7.47, 8.28, 9, 9.75
**47**	1.08, 1.95, 2.76, 4.14, 5.34, 5.91, 6.51, 8.4, 9.9
**48**	0.96, 1.53, 2.43, 3.06, 3.54, 4.35, 4.98, 5.64, 7.8, 9.57, 10.65
**49**	1.2, 1.83, 2.34, 3.27, 4.02, 4.83, 6.36, 6.99, 8.25, 8.97, 10.32
**50**	6.36, 7.56, 8.4, 8.82, 9.48, 9.87
**51**	1.23, 2.61, 4.56, 5.94, 7.74, 9.12, 9.84
**52**	1.2, 2.28, 2.91, 4.41, 5.31, 7.2, 8.19, 9.36
**53**	0.81, 1.77, 2.7, 3.27, 4.11, 5.04, 5.67, 6.21, 7.47, 8.43, 9.48
**54**	0.93, 1.59, 2.16, 2.67, 3.48, 4.56, 5.28, 6.33, 6.96, 7.86, 9.03, 9.87
**55**	3.39, 3.93, 4.62, 5.4, 8.73
**56**	2.76, 3.42, 4.2, 4.74, 5.19, 6.3, 7.44, 8.46, 8.88, 9.81
**57**	2.04, 2.49, 3.21, 4.23, 5.91, 6.57, 9, 9.81
**58**	4.41, 5.46, 6.06, 7.59, 8.1, 8.67, 9.66
**59**	0.93, 1.83, 2.7, 3.57, 4.38, 5.01, 5.64, 6.24, 7.2, 8.76
**60**	1.02, 1.95, 3.03, 3.66, 4.17, 4.89, 5.25, 6.63, 8.1, 8.79, 9.51
**61**	1.68, 2.22, 2.88, 3.69, 4.32, 4.71, 6.21, 10.02
**62**	2.1, 9.48
**63**	0.66, 1.26, 2.31, 3.27, 3.9, 4.89, 5.4, 5.85, 6.676, 7.56, 8.67, 9.42
**64**	2.01, 2.97, 3.63, 4.17, 5.13, 5.88, 6.81, 7.83, 8.55, 9.36, 9.9
**65**	1.05, 1.98, 3.54, 4.05, 5.49, 6, 6.57, 7.38, 7.92, 8.76, 9.69
**66**	1.02, 1.86, 2.64, 3.18, 4.38, 4.95, 5.91, 6.57, 9.12, 9.63
**67**	0.78, 1.47, 3.18, 3.72, 4.41, 5.88, 6.84, 8.79, 9.96
**68**	0.84, 2.52, 3.06, 4.2, 4.71, 6.18, 6.51, 7.38
**69**	1.77, 2.25, 3.09, 3.63, 4.86, 5.43, 6.12, 6.84, 7.74, 8.94, 10.05
**70**	0.9, 1.65, 2.43, 3.45, 4.53, 5.61, 6.45, 7.62, 8.67, 9.6, 10.62
**71**	0.93, 1.8, 2.46, 3.51, 4.32, 5.34, 6.3, 7.35, 7.89, 8.46, 9.24, 9.57, 10.41
**72**	0.81, 2.07, 4.47, 5.19, 6, 6.84, 7.86, 9.09, 9.6, 10.86
**73**	0.81, 2.31, 3.161, 3.78, 4.83, 5.67, 6.54, 7.59, 7.92, 8.58, 9.9, 10.41
**74**	2.43, 3.48, 4.14, 4.62, 5.19, 5.79, 6.12, 7.35, 7.83, 8.61, 9.54, 10.02
**75**	0.69, 2.34, 2.88, 4.11, 4.71, 6.66, 8.55, 9.06, 9.45, 10.92
**76**	0.96, 1.596, 1.89, 4.08, 4.74, 5.37, 6, 6.75, 8.79, 9.48, 10.29
**77**	0.9, 1.59, 2.19, 2.55, 3.84, 4.41, 5.28, 6.54, 7.11, 8.46, 9.57, 10.8
**78**	0.84, 2.49, 3, 3.96, 5.64, 5.97, 6.51, 8.37, 9.69, 10.32
**79**	0.96, 1.68, 3.03, 3.99, 5.94, 6.93, 8.01, 9.01, 9.96, 10.44
**80**	2.07, 4.38, 5.46, 6.54, 7.8, 8.22, 9.12, 9.78, 10.23, 10.74
**81**	0.93, 1.71, 3.03, 4.65, 5.28, 5.97, 8.49
**82**	0.84, 1.59, 2.16, 3.15, 3.72, 4.32, 5.46, 6.51, 7.71, 9.42
**83**	0.84, 1.44, 1.95, 2.49, 3.48, 4.05, 4.56, 5.46, 6.45, 6.96, 7.53, 8.64, 9.45, 10.47
**84**	0.9, 1.56, 4.14, 5.13, 5.85, 6.78, 7.2, 8.01, 10.77
**85**	0.84, 3.21, 3.72, 6.27, 6.75, 8.31, 8.91
**86**	1.68, 2.58, 3.39, 4.53, 5.64, 5.94, 6.9, 7.77, 9.21, 10.02
**87**	0.78, 2.04, 3.87, 4.74, 6.57, 7.11, 7.68, 8.28, 8.88, 9.33, 10.2
**88**	2.28, 3.3, 3.66, 4.35, 4.83, 5.61, 8.43, 9.93
**89**	0.57, 1.11, 1.65, 2.4, 4.08, 4.71, 5.46, 6, 6.87, 7.89, 9.24, 10.38
**90**	0.96, 1.83, 2.55, 3.09, 3.96, 4.8, 6, 7.44, 8.37, 9.57
**91**	0.72, 1.26, 1.86, 2.4, 2.94, 3.54, 4.11, 5.61, 6.3, 6.93, 7.56, 8.52, 8.94, 9.51, 9.99
**92**	0.63, 1.23, 1.83, 2.46, 2.97, 3.51, 4.47, 4.89, 5.97, 6.84, 7.35, 8.13, 9.33, 9.63, 10.86
**93**	1.5, 2.61, 3.45, 4.14, 4.71, 5.88, 7.5, 8.58, 9.81
**94**	0.9, 1.71, 2.46, 2.79, 3.42, 5.04, 6.57, 7.17, 7.27, 9.18, 10.29
**95**	1.56, 2.6, 4.95, 5.67, 6.54, 7.38, 8.25, 8.94, 9.6, 10.11
**96**	1.2, 2.82, 7.29, 7.98, 8.4, 9.24, 9.9, 10.65
**97**	0.75, 1.98, 2.7, 3.12, 3.75, 4.92, 6.06, 6.72, 7.2, 7.83, 8.61, 9.45, 10.26
**98**	0.75, 1.26, 1.98, 2.76, 3.24, 4.32, 5.97, 6.69, 7.5, 9.33, 10.08
**99**	4.11, 5.1, 5.55, 6.15, 6.75, 7.5, 8.67, 9.72
**100**	2.22, 2.82, 3.12, 4.83, 5.19, 6.81, 7.65, 8.64, 9.66, 10.35

**Table 3 biomimetics-08-00531-t003:** Antenna elements based on the leaf geometries that operate between 0.6 GHz and 1 GHz.

1		3	4	5		7		9	10
11	12	13		15	16		18	19	20
	22		24	25	26	27	28	29	30
31			34	35	36	37	38	39	40
	42	43	44	45	46	47	48	49	
	52	53	54		56			59	60
61		63		65	66	67	68		70
71	72	73	74	75	76	77	78	79	80
81	82	83	84	85		87	88	89	90
91	92	93	94	95	96	97	98		100

**Table 4 biomimetics-08-00531-t004:** Antenna elements based on the leaf geometries that operate in the L band (1–2 GHz).

1	2	3		5				9	10
	12	13		15	16		18	19	20
			24	25	26	27			
					36		38	39	40
	42	43	44	45	46	47	48	49	
51	52	53	54					59	60
61		63		65	66	67		69	70
71					76	77		79	
81	82	83	84		86			89	90
91	92	93	94	95	96	97	98		

**Table 5 biomimetics-08-00531-t005:** Antenna elements based on the leaf geometries that operate in the S band (2–4 GHz).

1	2	3	4	5		7	8	9	10
11	12	13		15	16	17	18	19	20
	22	23	24	25	26	27	28	29	30
31		33	34	35	36	37		39	40
41	42	43	44		46	47	48	49	
51	52	53	54	55	56	57		59	60
61	62	63	64	65	66	67	68	69	70
71	72	73	74	75		77	78	79	80
81	82	83		85	86	87	88	89	90
91	92	93	94	95	96	97	98		100

**Table 6 biomimetics-08-00531-t006:** Antenna elements based on the leaf geometries that operate in the C band (4–8 GHz).

1	2	3	4	5	6	7	8	9	10
11	12	13	14	15	16	17	18	19	20
	22	23	24	25	26	27	28	29	30
31	32	33	34	35	36	37	38	39	40
41	42	43	44	45	46	47	48	49	50
51	52	53	54	55	56	57	58	59	60
61		63	64	65	66	67	68	69	70
71	72	73	74	75	76	77	78	79	80
81	82	83	84	85	86	87	88	89	90
91	92	93	94	95	96	97	98	99	100

**Table 7 biomimetics-08-00531-t007:** Antenna elements based on the leaf geometries that operate in the X band (8–12 GHz).

1	2	3	4		6		8	9	10
11	12	13	14	15	16	17	18	19	20
21	22	23	24	25	26	27	28	29	30
31	32	33	34	35	36	37	38	39	40
41	42	43	44	45	46	47	48	49	50
51	52	53	54	55	56	57	58	59	60
61	62	63	64	65	66	67		69	70
71	72	73	74	75	76	77	78	79	80
81	82	83	84	85	86	87	88	89	90
91	92	93	94	95	96	97	98	99	100

**Table 8 biomimetics-08-00531-t008:** Antenna elements based on the leaf geometries that operate at frequencies of 2.4 GHz and 5 GHz.

Frequency	Leaf Geometries or Antenna Elements				
**2.4 GHz**	9	10	13	23	25
	28	39	42	46	49
	57	63	70	71	73
	74	75	78	83	89
	91	92			
**5 GHz**	7	8	17	18	33
	38	44	53	59	66
	95	97	99		
**Both**	26	31	48	94	

**Table 9 biomimetics-08-00531-t009:** Comparative analysis of the performance of each plant leaf design.

Plant Leaf or Antenna Element	Geometry	Design Impact	Operation Frequencies below 10 GHz (GHz)
*Tetrapterys macrocarpa* Malpighiaceae	elliptic	six frequency bands below 5 GHz	0.87, 1.62, 2.19, 2.61, 3.3, 4.11, 9.27
*Quercus alba* Fagaceae	pinnately lobed	widest frequency band	2.49, 3.18, 3.84, 4.62, 5.88, 7.65, 8.49, 9.54
*Cissampelos owariensis* Menispermaceae	circular base	better radiation characteristics	1.2, 1.71, 2.49, 3.06, 3.87, 6.27, 8.73, 11.61
*Liquidambar styraciflua* Hamamelidaceae	lobed leaves	seven frequency bands below 10 GHz	2.01, 4.23, 5.58, 6.06, 7.35, 7.92, 8.37, 9.09, 9.57, 10.83
*Sarcoshachis naranjoana* Piperaceae	ovate	eight frequency bands below 10 GHz	1.26, 2.34, 3.24, 3.63, 4.65, 5.43, 6.75, 7.26, 7.8, 8.36, 8.91, 9.51, 10.02
*Paranomus spectrum* Proteaceae	obovate	best behavior for high frequencies	1.59, 2.34, 2.76, 5.31, 5.94, 6.48, 7.56, 8.7, 9.84
*Cesearia ilicifolia* Salicaceae	toothed	more freq. bands (below 10 GHz)	0.9, 1.74, 2.46, 3.3, 4.62, 5.91, 6.51, 6.99, 7.47, 8.28, 9.00, 9.75

## Data Availability

The data presented in this study are available on request from the corresponding author.

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
