# Peer review of "Plant Foliar Geometry as a Biomimetic Template for Antenna Design"

_biomimetics, 2023, doi:10.3390/biomimetics8070531_

Round 1
Reviewer 1 Report
Comments and Suggestions for Authors
Authors analyzed the foliar geometries of 100 plant species, and applied them as a biomimetic design template for microstrip patch antenna systems. I think, paper is interesting, my minor suggestions are as follows:
1. Font size should be increased in the graphs in Figures 2-8.
2. Authors should enhance the analysis of Figure 2-8 in the manuscript.
3. Authors are missing some recent articles in the field related to the antenna design such as Tunable Plasmonic Properties and Absorption Enhancement in Terahertz Photoconductive Antenna Based on Optimized Plasmonic Nanostructures.
4. Authors should clarify how SolidWorks was used in order to process images.
Author Response
Response to Reviewer 1 Comments
Authors analyzed the foliar geometries of 100 plant species, and applied them as a biomimetic design template for microstrip patch antenna systems. I think, paper is interesting, my minor suggestions are as follows:
1. Font size should be increased in the graphs in Figures 2-8.
Response: The font size was increased in the graphs of Figures 2-8. Please see the revised version of our paper.
2. Authors should enhance the analysis of Figure 2-8 in the manuscript.
Response: The analysis of Figures 2-8 was enhanced in our manuscript. Please see pages 12-13 of the revised version of our paper.
3. Authors are missing some recent articles in the field related to the antenna design such as Tunable Plasmonic Properties and Absorption Enhancement in Terahertz Photoconductive Antenna Based on Optimized Plasmonic Nanostructures.
Response: Our manuscript was corrected to include that reference and others more related to the antenna design. Please see the revised version of our paper.
4. Authors should clarify how SolidWorks was used in order to process images.
Response: SolidWorks processes the image in black and white. The function “Sketch picture” of SolidWorks can be used to detect the black and white leaf geometry. Then, a collection of samples or points are generated to provide a tridimensional model by using a compatible format with CST solver. This comment was added to the revised version of our paper (see page 3).
Thanks for your comments!

Reviewer 2 Report
Comments and Suggestions for Authors
1. Please provide quantitative results in the abstracts
2. The literature review conducted in this paper is not thorough enough. More related papers should be covered to explain the development of this research area.
3. While the problem statements of current work has been clearly explained, it is also crucial for authors to highlight the main differences of their work with the existing ones to highlight their uniqueness.
4. The work flow proposed in Figure 1 is quite brief. A more comprehensive version of work flow is needed to provide more detailed explanation on the proposed methodology.
5. The results in Figures 2 to 8 are not clearly illustrated because they are too many subfigures are squeezed into a main figure. Please enhance the readability.
6. Please explain the limitations of current work and the potential future works that can be extended from current study.
Comments on the Quality of English LanguageNo major issues with the English language quality.
Author Response
Response to Reviewer 2 Comments
1. Please provide quantitative results in the abstract.
Response: The abstract was corrected to provide quantitative results. Please see the abstract of revised version of our paper.
2. The literature review conducted in this paper is not thorough enough. More related papers should be covered to explain the development of this research area.
Response: Our manuscript was corrected to include more related papers in the research area. Now, the literature review is more complete. Please see the revised version of our paper.
3. While the problem statements of current work has been clearly explained, it is also crucial for authors to highlight the main differences of their work with the existing ones to highlight their uniqueness.
Response: Our manuscript was corrected to highlight the main differences with respect to previous work. Please see page 2 of the revised version of our paper.
4. The work flow proposed in Figure 1 is quite brief. A more comprehensive version of work flow is needed to provide more detailed explanation on the proposed methodology.
Response: The methodology set in Figure 1 was explained in a more comprehensive version providing more details. Please see page 3 of the revised version of our paper.
5. The results in Figures 2 to 8 are not clearly illustrated because they are too many subfigures are squeezed into a main figure. Please enhance the readability.
Response: Figures 2-8 was enhanced to be clearly illustrated in our manuscript. Furthermore, the analysis of these figures was enhanced. Please, see Figures 2-8 and pages 12-13 of the revised version of our paper.
6. Please explain the limitations of current work and the potential future works that can be extended from current study.
Response: Some limitations of this work and potential future work can be extended from current study as follows:
- Each leaf architecture (of this work) considers vein characteristics of first order. Then, more vein characteristics (of higher order) can be considered in order to obtain better performance features.
- This study only considers the response of the leaf geometry found in nature. It could be interesting try to generate a geometry that considers the optimization of the design parameters in order to improve performance. Furthermore, some of the interesting characteristics found in this study could be used (or mixed) to try to create new design geometries or super-geometries. Therefore, we consider that future work searching the optimization of biomimetic designs based on plant leaf architectures will further support the development of specific applications in antenna design.
This was added to the revised version of our paper (see page 18).
All the comments helped a great deal to improve the quality of our paper.
Thanks!
